# Chronological Age, Somatic Maturation and Anthropometric Measures: Association with Physical Performance of Young Male Judo Athletes

**DOI:** 10.3390/ijerph18126410

**Published:** 2021-06-13

**Authors:** Bruno B. Giudicelli, Leonardo G. O. Luz, Mustafa Sogut, Hugo Sarmento, Alain G. Massart, Arnaldo C. Júnior, Adam Field, António J. Figueiredo

**Affiliations:** 1Faculty of Sport Science and Physical Education, Research Unit for Sport and Physical Activity (CIDAF), University of Coimbra, 3004-531 Coimbra, Portugal; leonardoluz@arapiraca.ufal.br (L.G.O.L.); hugo.sarmento@uc.pt (H.S.); alainmassart@fcdef.uc.pt (A.G.M.); afigueiredo@fcdef.uc.pt (A.J.F.); 2Kinanthropometry, Physical Activity and Health Promotion Laboratory (LACAPS), Federal University of Alagoas—Campus Arapiraca, Arapiraca 57309-005, Brazil; arnaldo.junior@arapiraca.ufal.br; 3Department of Physical Education and Sports, Faculty of Education, Middle East Technical University, Ankara 06800, Turkey; msogut@metu.edu.tr; 4School of Human and Health Sciences, University of Huddersfield, Huddersfield HD1 3DH, UK; adam.field@hud.ac.uk

**Keywords:** adolescent athlete, combat sports, body composition, bio-banding

## Abstract

Sport for children and adolescents must consider growth and maturation to ensure suitable training and competition, and anthropometric variables could be used as bio-banding strategies in youth sport. This investigation aimed to analyze the association between chronological age, biologic maturation, and anthropometric characteristics to explain physical performance of young judo athletes. Sixty-seven judokas (11.0–14.7 years) were assessed for anthropometric and physical performance. Predicted adult stature was used as a somatic maturation indicator. A Pearson’s bivariate correlation was performed to define which anthropometric variables were associated with each physical test. A multiple linear hierarchical regression was conducted to verify the effects of age, maturity, and anthropometry on physical performance. The regression models were built with age, predicted adult stature, and the three most significantly correlated anthropometric variables for each physical test. Older judokas performed better in most of the physical tests. However, maturation attenuated the age effect in most variables and significantly affected upper body and handgrip strength. Anthropometric variables attenuated age and maturity and those associated with body composition significantly affected the performance in most tests, suggesting a potential as bio-banding strategies. Future studies should investigate the role of anthropometric variables on the maturity effect in young judokas.

## 1. Introduction

The complex process of growth and maturation must be considered for children and adolescents in sport to ensure suitable training and competition routines. Chronological age is the traditional strategy to categorize young athletes appropriately for their level of development [1]. While growth is the process of increasing body size in whole or in parts, biological maturation refers to physiological and cognitive development towards adulthood. Although maturational events have an established order in which they happen, the moment when they occur and their duration have immense variability between individuals, even at the same age, which affect the physical, technical, and psychological performance of young athletes. This happens more prominently in boys between 13 and 16 years old [2], which may increase the risk of injury [3] and impair motivation due to the performance discrepancy [4], influencing whether the young athlete will continue in sports practice long-term [5]. In sports where strength, power, and speed are paramount, and in those where physical contact is inevitable, mature individuals tend to have a physical advantage over their less mature peers, since young people that mature and develop early tend to be taller and heavier [6].

Several investigations have been carried out to examine the effect of growth and maturation on young athletes’ performance and to seek alternative strategies to chronological age for the categorization of young athletes, namely those that are based on the use of body size and/or maturational status [7,8,9]. These strategies are called bio-banding and do not disregard other aspects that must be considered regarding the allocation of young athletes in competitive categories, such as skill level and psychological profile [10]. However, most of these investigations focus on team sports, mainly in soccer, some with support from official sporting entities, and have already resulted in the first experiences of unofficial tournaments using bio-banding to distribute young athletes in competitive categories, with positive results [11]. In combat sports, less research attention has been given to the plausible effect of maturation over performance [12,13], and to the applicability of bio-banding to different modalities. Nonetheless, grouping young combat athletes based on physical attributes is common (e.g., boxing, judo, taekwondo, wrestling). In these modalities, athletes are grouped based on chronological age and body mass, and compete in weight classes to promote fair competition and reduce potential injuries [14]. Although studies on the topic are still scarce [13], there is evidence of maturation effect within weight categories in young combat sports [15], raising questions about the suitability of body mass as criteria to guarantee equal conditions among athletes, justifying research on the topic.

In addition, various research investigated the use of anthropometric variables for detection and prediction of success in young athletes [16], which can potentially also be used as bio-banding alternatives instead of body mass for categorizing young athletes in various sports, including combat sports. Based on the above-mentioned factors, the aim of the present investigation was to analyze the association between chronological age, biological maturation, and anthropometric characteristics to explain the physical performance of young judo athletes. Assuming that anthropometric characteristics may mitigate the effect of chronological age and biological maturation on the performance of young judo athletes, bio-banding strategies in judo and other combat sports could be developed using anthropometric variables, aiming to promote training and competition routines best suited to the development stages of young combat sport athletes.

## 2. Materials and Methods

### 2.1. Participants

This is a cross-sectional study with a convenience sample, consisting of 67 young male judokas aged 11.0–14.7 years old selected from eight clubs in Portugal. To be included in the study, the participants needed to be between 11.0 and 14.9 years old, have at least one year of judo training, and have no physical or psychological contraindications to participation. Parents or legal guardians provided signed informed consent prior to data collection. Verbal consent was also obtained from participants. Two participants from different clubs dropped out of the study once data collection had commenced. The study was conducted in accordance with the Declaration of Helsinki for Human Studies of the World Medical Association and approved by the Ethics Committee of the Faculty of Sports Sciences and Physical Education of the University of Coimbra [CE/FCDEF-UC/00452019].

### 2.2. Anthropometric Measures

Common anthropometric procedures [17] were adopted. Stature and sitting height (SH) were measured using a portable stadiometer (Seca Bodymeter 206; Seca Deutschland, Hamburg, Germany) and a segmometer (Rosscraft, T.E. and B. Ross, Perth, Australia), respectively. The inferior members’ length (IML) was estimated as stature minus SH. Arm span (AS) was measured using a metallic anthropometric tape by assessing the distance between right and left dactylion points, with both arms abducted 90 degrees. The hand length (HL) was measured as the distance between the stylion and dactylion, while the foot length (FL) was measured as a straight distance between the acropodion and pterion points using an anthropometer. Arm circumference (AC) and calf circumference (CC) were measured with a metallic anthropometric tape. All measures were taken to the nearest 0.1 cm. Body mass (BM) was measured to the nearest 0.1 kg using a portable digital scale (Seca Bella 840; Seca Deutschland, Hamburg, Germany). Skinfold thickness was assessed to the nearest 0.1 mm using a Rosscraft skinfold calipers in the following references: triceps, subscapular, suprailiac, and calf. Estimates of fat mass percentage were obtained from the sex-specific equation derived from the sum of the triceps and subscapular skinfolds [18]. Thereafter, estimated body fat mass (BFM) and body fat-free mass (BFFM) were calculated.

### 2.3. Biological Maturation

Predicted adult stature (PAS) was used as a maturational indicator [19]. It has been adopted in investigations on the biological maturation effect on physical fitness of children in general [20] and in research focusing on youth performance in sport [21], due to the feasibility compared with more valid but invasive indicators [22]. Predicted adult stature has been used instead of peak of height velocity (PHV) in studies on bio-banding, since both have correspondence with pubertal status given by stage of pubic hair [10], but the latter seems to have limited application depending on chronological age and actual age at the peak of height velocity [23]. The PAS protocol requires the participants’ decimal age, stature, and body mass, as well as the average parental stature. The stature of the parents was collected through a questionnaire attached to the informed consent form. The PAS variable was also expressed as the percentage of predicted adult stature attained (APAS). It is assumed that among children of the same chronological age, individuals with higher APAS are more advanced in somatic maturation compared with individuals with lower APAS [2].

### 2.4. Physical Performance

The pacer test was used to evaluate aerobic performance, with the number of completed laps being used as performance indicator. Agility was measured using a 10 × 5 m shuttle-run test, with the time to complete all laps recorded in seconds [24]. The line-drill test was used to evaluate anaerobic performance, with the time taken to complete the course expressed in seconds [25]. Strength was assessed using the following indicators: abdominal muscle strength (AMS), 60-s sit-up test [26]; upper body muscle strength (UBS), 2-kg medicine ball throw [27]; lower body muscle strength (LBS), standing long jump test [24]; and dominant-hand grip strength (HgS), measured by a dynamometer (Lafayette) [24]. The best of two attempts was recorded in kilograms for HgS.

### 2.5. Procedures

All data were collected by the same trained team, in a single visit for each judo club, where the anthropometric measurements were carried out initially, followed by the physical performance assessments. Participants completed a warm-up, under the guidance of a trainee researcher, before each station was completed in circuit form, in the following order: (1) pacer; (2) 2 kg standing medicine ball throw; (3) stand broad jump test; (4) 10 × 5 m shuttle-run test; (5) sit-ups; (6) handgrip strength; and (7) line-drill test.

### 2.6. Statistical Analysis

Descriptive statistics (ranges, means, standard deviations, and 95% confidence intervals) were used for chronological age (CA), APAS, anthropometric characteristics, and physical test performance. The Kolmogorov–Smirnov test was used to test normality: body fat mass, agility, lower body strength, and handgrip strength were significant. A Pearson’s bivariate correlation, with 95% bias corrected and accelerated confidence limits based on 1000 bootstrap samples, to correct data normality [28], was performed to assess the level of association between all variables. Multiple linear regression, which does not assume the assumption of data normality [28], was conducted to verify the influence of the maturity indicators and anthropometric variables on physical performance. CA and APAS were previously selected for the first and second regression models, respectively, based on their impact on the performance of young athletes described in the literature [2]. For each performance test, the three anthropometric variables with the highest correlation coefficient were selected for the third model. Independent variables were inserted into the regression models hierarchically: Model 1 was constituted by CA; Model 2 by CA and APAS; and Model 3 by CA, APAS, and the three anthropometric variables with the highest correlation coefficients for each physical performance test. Most models met all assumptions for the multiple linear regression [28]: error independence (Durbin-Watson values between 1–3), non-multicollinearity (Tolerance values >0.1; VIF values <10), homoscedasticity (standardized residual values between −3 and 3) and non-influential cases (Cook’s distance values <1). The exception was LBS, which failed on the assumption of non-influential cases. Significance of *p* < 0.05 was adopted in the analyses. IBM SPSS 26.0 software (SPSS, Inc., Chicago, IL, USA) was used in the study.

## 3. Results

Table 1 presents the descriptive statistics for the total sample, showing ranges, means, standard deviations, and 95% confidence intervals for CA, APAS, anthropometric characteristics, and physical test performance.

Table 2 summarizes the Pearson’s bivariate correlation coefficient between maturational indicators and anthropometric variables, and the physical performance. Significant moderate to high correlations were found for CA and APAS with the performance variables, except for agility, in which only CA was significant, and except for abdominal strength. Among the measured anthropometric variables, body fat mass correlated with all physical tests associated with running, and with abdominal strength. Body fat-free mass correlated with the neuromuscular strength performance, except for abdominal strength. Body mass, arm circumference, and calf circumference were not selected for any regression model, as no significant correlations were found.

Table 3 showed the multiple linear regression models for the aerobic, anaerobic and agility physical test performances. Model 1 significantly explained 16.8% of the aerobic performance, 19.9% of the anaerobic performance, and 6.2% of the agility performance. The effect of age on these tests was mitigated with the inclusion of the somatic maturation indicator. Model 2 was not significantly associated with the agility test since there was no significant correlation between agility and APAS. Model 3 explained the performance relative to aerobic (44.6%), anaerobic (47.6%), and agility (24.3%) tests significantly better than the other models. The inclusion of the anthropometric variables attenuated the effects of CA and APAS. Body fat mass (β = −0.507, *p* < 0.001) and hand length (β = 0.262, *p* = 0.046) explained aerobic performance better than the other variables. For anaerobic performance, only body fat mass (β = 0.536, *p* < 0.001) was highlighted with a significantly effect. For agility test performance, body fat mass (β = 0.360, *p* = 0.003), and superior members length (β = −0.290, *p* = 0.034) emerged as being significantly associated.

The regression models for neuromuscular strength were presented in Table 4. Model 1, containing only CA, explained lower body strength better than other models. Model 2 was better suited for UBS (46.3%) and HgS (44.4%) to explain performance than Model 1. With its inclusion in Model 2, APAS attenuated the CA effect, while remaining significant (β = 0.544, *p* = 0.008 for UBS; β = 0.897, *p* < 0.001 for HgS). Model 3 explained performance better in the majority of the neuromuscular strength tests, except for LBS. The addition of the anthropometric variables attenuated age and maturity effects and significantly explained 23.9% of AbS, 71.6% of the UBS, and 79% of the HgS. Body fat mass better explained AbS (β = −0.295, *p* < 0.05), while age (β = 0.402, *p* < 0.011) and body fat-free mass (β = 0.747, *p* < 0.001) better explained UBS. Only body fat-free mass was significantly related to HgS (β = 0.626, *p* < 0.01). Superior members length was the only variable significantly related to LBS (β = 0.318, *p* < 0.05).

## 4. Discussion

The purpose of this investigation was to analyze the interaction between chronological age, somatic maturation, and anthropometric characteristics to explain the performance in young judo athletes. Chronological age had a significant effect on all physical tests, with older judokas performing better, except on abdominal strength. However, somatic maturation attenuated its effect in most of the tests (aerobic and anaerobic performances, upper body, lower body, and handgrip strengths) and its main effect was significant and positive on upper body and handgrip strengths. Furthermore, regression models with anthropometric variables better explain physical performance since chronological age and maturation significance decreased when anthropometric variables were included in the model. Among them, body fat mass was negatively related to tests associated with running (aerobic, anaerobic and agility tests), and to abdominal strength, while chronological age and body fat-free mass were positively related to upper body strength, and only body fat-free mass to handgrip strength.

Judo is characterized by intermittent strength and power actions interspersed with moments of recovery. The athlete, through grappling techniques, seeks to throw the opponent to the ground, whereby the contest occasionally continues until one of the athletes is immobilized or surrenders [29]. According to the results, maturity status could be deemed as having greater importance than chronological age in relation to the performance of young male judokas aged 11–15. Upper body and handgrip strength are considered critical to successful judo performance, since the sport involves pulling and pushing the opponent to allow the application of throwing, immobilizing, and blocking techniques [30,31]. However, current literature provides limited evidence regarding the influence of maturity on muscular strength in young judo athletes. One of the few studies on the subject, which involved testing a sample similar to the present study (66 young male judokas, mean age: 13.9 years), found that somatic maturation, along with body mass and stature, significantly explained handgrip strength, but not upper body strength [32]. Comparable results have also been observed in previous studies conducted on young male athletes from different sports [33,34,35].

Maturity-related differences in medicine ball throw and handgrip strength were observed, after age control, in young basketball players aged 12–13 years [33]. Significant differences between contrasting maturity groups in terms of handgrip strength were evidenced among 14 year old handball players [34]. In support of the present research, another investigation found significant and positive correlations, when controlling for chronological age, between maturity and handgrip strength and various medicine ball throw tasks in young tennis players aged 8–16 years [35].

Inter-individual differences in maturity can either positively or negatively affect performance in neuromuscular tests. Additionally, the nature of the association may vary depending on the age and sex of the individual, in addition to the characteristics of the task [36,37]. One of the first studies relating to biological maturation, growth variables, and physical tests performance suggested that while maturation of the neuromuscular system may contribute positively to the development of motor skill, maturity-related changes in both size and body composition could also negatively affect performance, particularly on tests requiring body displacement [36]. In the present study, body composition had a significant impact on performance. Body fat mass significantly explained aerobic, anaerobic, and agility performance, with heavier judokas performing worse. On the other hand, body fat-free mass significantly and positively explained superior limb strength (upper body and handgrip strengths tests), which are variables most associated to the maturity effect. The importance of body composition, namely with low fat mass and high fat-free mass, is well-established as being a determinant of successful judo performance [38,39,40]. However, grouping young judokas into weight categories using total body mass disregards body composition, which can vary significantly between weight categories [41], and seems to be relevantly related to biological maturation, as already been observed in a previous study [42], and evidenced in the present investigation. These findings may reinforce doubts about the effectiveness of the young athlete’s distribution in weight categories for equalization of competition conditions and training routines, evidencing the need to control the effect of maturation in judo with/bio-banding strategies, and raising the possibility of using other anthropometric characteristics as a categorization criterion, such as the APAS itself or components of body composition.

Previous studies have investigated the relationship between morphological characteristics, biological maturation, and motor performance in young judo athletes using regression analysis methods [32,43,44]. However, few studies have adopted the hierarchical procedure in regression analyses, such as the present study. This involved the current research team deciding which variables should enter first in the regression models, based on previous knowledge, with such an approach considered more adequate compared with stepwise methods [28]. However, there are some limitations that should be considered for the present investigation. The lack of power analyses to determine whether the sample size was adequate may limit statistical inferences. The use of indirect tests to measure physical capacity may also limit the investigation. Notably, fat mass was significantly associated to judokas performance in running tests, which involve displacement of total body mass, with heavier judokas having presented significantly lower performance. The Special Judo Fitness Test (SJFT) is an alternative widely used in the literature for the physical evaluation of judo athletes, but in addition to also involving running in its execution, there is still no consensus on its discriminating power in the young population [45]. In several studies that involve the use of APAS as a maturational indicator, including this one, the statures of the participants’ parents were verified by self-reporting, a procedure that has support in the literature [22] but might introduce bias [46]. The error between predicted and actual mature stature at 18 years of age is reported to be 2.1% [19]. As explained earlier, predicted adult stature has been used instead of peak of height velocity in studies on bio-banding, since the latter seems to have limited application depending on age at the peak of height velocity for both boys and girls [47]. Considering the above, the present study used APAS to avoid the impact of the wide range of age at the peak of height velocity promoted by the biological variability at this auxological period. Moreover, the predicted adult stature is being used in most of the studies related with the bio-banding approach and this can allow the present study to be more objective compared with further research in the literature.

## 5. Conclusions

The results of the present study corroborate the findings of previous research and add value not only because they support the significant effect of the somatic maturation on young judo athletes’ physical performance, mainly on upper body and handgrip strengths tests. These findings suggest that bio-banding may be an effective strategy to reduce the implications of maturation on judo performance, preventing those that are less mature being at a physical disadvantage, assisting trainers in the construction of training routines appropriate to the youth’s development, supporting sporting entities in proposing fairer tournaments, reducing injury risk, and promoting engagement in sports for the long-term. Additionally, anthropometric characteristics, such as body fat mass and body fat-free mass, may need to be considered regarding biological maturational effect. Future studies should assess bio-banding strategies for categorizing young judo athletes and other combat sports, in the search for more appropriate training and competition conditions. This may promote positive sporting experiences for young athletes, precluding unfair disadvantages and improving performance. Other research approaches may involve further investigating the role of anthropometric variables over the maturation effect on young judo athletes controlling for age and body mass (the traditional criteria for bio-banding in combat sports). Furthermore, an avenue for future research may be to evaluate the impact of participating in bio-banding tournaments on the performance and on the quality of the experience perceived by young judo and other combat sport athletes.

## Figures and Tables

**Table 1 ijerph-18-06410-t001:** Descriptive statistics for the total sample (*n* = 67).

Variables	Range		Mean		*sd*
	Minimum	Maximum	Value	95%CI	
Chronological age (years)	11.01	14.70	12.54	12.30 to 12.78	0.99
APAS (%)	77.0	94.0	84.4	83.2 to 85.5	4.7
Body mass (kg)	27.6	79.6	47.6	44.7 to 50.5	11.2
Body fat mass (kg)	2.1	34.4	9.6	8.0 to 11.1	6.3
Body fat-free mass (kg)	25.5	65.1	38.0	36.1 to 39.9	7.8
Stature (cm)	134.8	176.5	154.0	151.6 to 156.4	9.9
Sitting height (cm)	71.5	93.2	80.0	78.8 to 81.2	5.1
Arm span (cm)	133.0	180.0	154.1	151.5 to 156.7	10.8
Superior members length (cm)	36.2	70.8	60.2	58.9 to 61.5	5.4
Hand length (cm)	14.1	21.3	16.9	16.5 to 17.2	1.5
Inferior members length (cm)	60.3	85.5	74.0	72.7 to 75.4	5.5
Foot length (cm)	20.1	29.0	24.4	24.0 to 24.9	2.0
Arm circumference (cm)	19.0	36.0	25.3	24.5 to 26.1	3.3
Calf circumference (cm)	27.0	40.1	32.6	31.8 to 33.4	3.3
Pacer test (m)	140	1740	757	680 to 835	318
Line-drill test (sec) *	30.09	46.60	36.14	35.36 to 36.92	3.20
Agility 10 × 5 shuttle run (sec) *	15.88	26.25	19.44	18.93 to 19.96	2.12
60-s sit-ups (count)	15	61	41	39 to 44	10
2-kg ball throw (m)	3.19	8.79	5.22	4.93 to 5.52	1.22
Standing long jump (m)	1.12	5.65	1.69	1.55 to 1.83	0.57
Hand grip strength (kgf)	14.0	40.0	24.8	23.4 to 26.2	5.8

95%CI, confidence interval; *sd*, standard deviation; APAS, attained predicted mature stature; * Runtime tests—lower value represents better performance.

**Table 2 ijerph-18-06410-t002:** Pearson bivariate correlation coefficients between maturational indicators and anthropometric variables, and physical tests performance variables, with 95% bias corrected and accelerated confidence limits based on 1000 bootstrap samples (*n* = 67).

Independent Variables	Dependent Variables (Physical Tests Performance)						
	Pacer Test (m)	Line-Drill test (s) ^+^	Agility 10 × 5 Shuttle Run (s) ^+^	60-s Sit-Ups (*n*)	2-kg Ball Throw (m)	Stand Long Jump (m)	Handgrip Strength (kg)
Maturational indicator							
Age	0.41 ** (0.21; 0.59)	−0.45 *** (−0.61; −0.24)	−0.25 * (−0.48; 0.02)	0.15 (−0.07; 0.37)	0.63 *** (0.42; 0.80)	0.47 *** (0.25; 0.67)	0.52 *** (0.29; 0.68)
APAS (%)	0.39 ** (0.18; 0.57)	−0.41 ** (−0.58; −0.21)	−0.19 (−0.43; 0.12)	0.06 (−0.20; 0.30)	0.68 *** (0.47; 0.83)	0.47 *** (0.24; 0.68)	0.65 *** (0.49; 0.77)
Anthropometry							
Body mass (kg)	−0.07 (−0.29; 0.19)	0.02 (−0.22; 0.21)	−0.10 (−0.14; 0.33)	−0.09 (−0.32; 0.17)	0.66 *** (0.48; 0.79)	0.10 (−0.14; 0.34)	0.66 *** (0.49; 0.78)
Body fat mass (kg)	−0.41 ** (−0.56; −0.19)	0.40 ** (0.16; 0.57)	0.32 ** (0.05; 0.53)	−0.27 * (−0.50; 0.03)	0.22 (0.01; 0.42)	−0.31 * (−0.50; −0.10)	0.26 * (0.01; 0.46)
Body fat-free mass (kg)	0.23 (0.04; 0.42)	−0.29 * (−0.46; −0.14)	−0.11 (−0.34; 0.10)	0.08 (−0.16; 0.32)	0.82 *** (0.71; 0.89)	0.40 ** (0.15; 0.61)	0.78 *** (0.66; 0.86)
Stature (cm)	0.25 * (0.06; 0.43)	−0.34 ** (−0.48; −0.17)	−0.19 (−0.40; 0.04)	0.05 (−0.22; 0.31)	0.71 *** (0.55; 0.82)	0.37 ** (0.16; 0.57)	0.73 *** (0.61; 0.82)
Sitting height (cm)	0.20 (−0.02; 0.40)	−0.25 * (−0.40; −0;07)	−0.14 (−0.31; 0.06)	0.05 (−0.20; 0.30)	0.69 *** (0.51; 0.82)	0.33 ** (0.13; 0.52)	0.71 *** (0.54; 0.82)
Arm span (cm)	0.16 (−0.02; 0.37)	−0.34 ** (−0.50; −0.18)	−0.21 (−0.43; 0.01)	0.11 (−0.16; 0.34)	0.73 *** (0.60; 0.82)	0.52 ** (0.13; 0.55)	0.70 *** (0.56; 0.81
Superior members length (cm)	0.13 (−0.08; 0.38)	−0.30 * (−0.54; −0.08)’	−0.26 * (−0.45; −0.10)	−0.01 (−0.23; 0.24)	0.58 *** (0.38; 0.78)	0.40 ** (0.19; 0.59)	0.56 *** (0.35; 0.79)
Hand length (cm)	0.38 ** (0.20; 0.55)	−0.31 * (−0.45; −0.16)	−0.15 (−0.35; 0.07)	0.31 * (0.10; 0.55)	0.55 *** (0.29; 0.74)	0.28 * (0.02; 0.53)	0.53 *** (0.29; 0.72)
Inferior members length (cm)	0.27 * (0.09; 0.44)	−0.38 ** (−0.53; −0.20)	−0.21 (−0.43; 0.04)	0.04 (−0.21; 0.29)	0.64 *** (0.48; 0.75)	0.36 ** (0.14; 0.55)	0.66 *** (0.55; 0.77)
Foot length (cm)	0.26 * (0.08; 0.46)	−0.22 (−0.40; −0.04)	−0.15 (−0.36; 0.09)	0.28 * (0.05; 0.51)	0.49 *** (0.26; 0.70)	0.21 (−0.01; 0.44)	0.50 *** (0.29; 0.69)
Arm circumference (cm)	−0.09 (−0.30; 0.13)	0.03 (−0.20; 0.24)	0.12 (−0.14; 0.36)	−0.05 (−0.28; 0.19)	0.66 *** (0.48; 0.78)	0.12 (−0.15; 0.35)	0.57 *** (0.37; 0.71)
Calf circumference (cm)	−0.15 (−0.37; 0.09)	0.06 (−0.18; 0.25)	0.16 (−0.09; 0.39)	−0.16 (−0.38; 0.12)	0.57 *** (0.38; 0.72)	0.01 (−0,22; 0.25)	0.53 *** (0.35; 0.68)

APAS attainted predicted adult stature; * *p* < 0.05; ** *p* < 0.01; *** *p* < 0.001; ^+^ Runtime tests—lower value represents better performance; BCa, bootstrap 95% CLs reported in brackets.

**Table 3 ijerph-18-06410-t003:** Hierarchical multiple linear regression models for physical performance tests (aerobic, anaerobic, and agility) in youth male judokas.

Physical Test	Multilinear Regression Models					
	Model	R^2^	*p*	Independent Variables	β	*p*
Pacer Test (m)	Model 1	0.168	0.001	Age	0.410	0.001 *
	Model 2	0.171	0.637	Age	0.307	0.215
				APAS	0.116	0.637
	Model 3	0.446	<0.001	Age	0.127	0.553
				APAS	0.254	0.330
				Body fat mass	−0.507	<0.001 *
				Hand length	0.262	0.046 *
				Inferior members length	−0.037	0.816
Line-drill test (s) ^+^	Model 1	0.199	<0.001	Age	−0.446	<0.001 *
	Model 2	0.200	0.753	Age	−0.379	0.120
				APAS	−0.076	0.753
	Model 3	0.476	<0.001	Age	−0.326	0.124
				APAS	0.040	0.876
				Body fat mass	0.536	<0.001 *
				Arm span	−0.267	0.152
				Inferior members length	−0.118	0.533
Agility 10 × 5 shuttle run (s) ^+^	Model 1	0.062	0.042	Age	−0.250	0.042 *
	Model 2 ^#^	-	-	Age	-	-
				APAS	-	-
	Model 3	0.243	0.002	Age	−0.365	0.145
				APAS	0.213	0.436
				Body fat mass	0.360	0.003 *
				Superior members length	−0.290	0.034 *

* *p* < 0.05; APAS attainted predicted adult stature; ^+^ Runtime tests—lower value represents better performance; ^#^ The models were not adequate to explain the variable.

**Table 4 ijerph-18-06410-t004:** Hierarchical multiple linear regression models for physical performance tests (neuromuscular strength) in youth male judokas.

Physical Test	Multilinear Regression Models					
	Model	R^2^	*p*	Independent Variables	β	*p*
60-s sit-ups (count)	Model 1 ^#^			Age		
	Model 2 ^#^			Age		
				APAS		
	Model 3	0.239	0.003	Age	0.359	0.147
				APAS	−0.442	0.102
				Hand length	0.296	0.236
				Foot length	0.157	0.493
				Body fat mass	−0.295	0.016 *
2-kg ball throw (m)	Model 1	0.399	<0.001	Age	0.632	<0.001 *
	Model 2	0.463	0.008	Age	0.150	0.450
				APAS	0.544	0.008 *
	Model 3	0.716	<0.001	Age	0.402	0.011 *
				APAS	−0.239	0.235
				Body fat-free mass	0.747	<0.001 *
				Arm span	0.250	0.130
				Stature	−0.238	0.210
Stand long jump (m)	Model 1	0.286	<0.001	Age	0.535	<0.001*
	Model 2	0.317	0.093	Age	0.199	0.374
				APAS	0.379	0.093
	Model 3	0.375	0.143	Age	0.274	0.233
				APAS	0.208	0.464
				Arm span	−0.213	0.336
				Body fat-free mass	0.114	0.615
				Superior members length	0.318	0.034 *
Handgrip Strength (kg)	Model 1	0.270	<0.001	Age	0.520	<0.001 *
	Model 2	0.444	<0.001	Age	−0.275	0.175
				APAS	0.897	<0.001 *
	Model 3	0.790	<0.001	Age	−0.044	0.805
				APAS	0.146	0.526
				Body fat-free mass	0.626	0.001 *
				Stature	0.194	0.435
				Sitting height	−0.113	0.632

^#^ The models were not adequate to explain the variable; APAS attainted predicted adult stature; * *p* < 0.05.

## Data Availability

The data presented in this study are available on request from the corresponding author. The data are not publicly available due to ethical restrictions.

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
