# Peer review of "Chronological Age, Somatic Maturation and Anthropometric Measures: Association with Physical Performance of Young Male Judo Athletes"

_ijerph, 2021, doi:10.3390/ijerph18126410_

Round 1

Reviewer 1 Report

General comments:

The study aimed to analyze the effects of chronological age, biological maturation, and anthropometric characteristics on the physical performance of young judo athletes. The subject is interesting and the results certainly can used by coaches and combat sports sports scientists. However, there is an issue that concerns me: the similarity with the study conducted by Giudicelli et al. (Bio-Banding in Judo: The Mediation Role of Anthropometric Variables on the Maturation Effect), recently published in IJERPH. The sample of both studies is the same (67 young male judokas aged 11.0–14.7 years), as we can see in the methods. In both studies the physical tests and anthropometric variables are also the same (see Table 1 in both studies). The predicted adult stature (APAS) is the same variable as predicted mature stature (PMS) (Giudicelli et al., 2020). The statistica analysis is very similar: the current study used Pearson correlation and multiple linear regression to analyze the relationship/prediction models, respectively, between anthropometric variables and physical tests. Giudicelli et al. (2020) used partial correlation (controlling for chronological age and body mass) and linear regression models between anthropometric variables and physical tests. Therefore, I don't have enough subsidies to classify the article as plagiarism, but I can say that there is a derived data analysis with very similar statistical analysis. I believe that the editor can decide on this issue.

As we express our gratitude for your review of our manuscript, we hereby present to the editors and reviewers the changes made to the manuscript, point-by-point, from the suggestions received. While we hope that the corrections made have met the demands arising from the review, we are at your disposal to clarify any doubts and make any other modifications that you may consider necessary to improve the quality of the manuscript.

This paper, along with two others, comes from my PhD in Sport Sciences and Physical Education at the University of Coimbra, which is nearing complete. The objective was to study the effect of biological maturation on the performance of young male judo athletes. For this reason, the three studies have the same sample and the same data collection, but with complementary analyses to understand the effect of biological maturation on the performance of judokas, something already stablished in team sports, but still with scarce production in combat sports.

The first study published in the IJERPH aimed to verify the effect of maturation on the judoka’s performance and the possibility of mediation of this effect by anthropometric variables. We concluded that there was a maturation effect over aerobic performance and handgrip strength, with body fat mass and body fat-free exerting total mediation on aerobic performance, and body fat mass, body fat-free mass, stature, arm span, and lower limbs length applying total mediation on handgrip strength.

On the present study, the objective was to confirm the maturational effect over the judoka’s performance, and to analyze the associated impact of age, maturity, and anthropometric characteristics through hierarchical regressions, very motivated by the recent study of Detanico D, Kons RL, Fukuda DH, Teixeira AS. Physical Performance in Young Judo Athletes: Influence of Somatic Maturation, Growth and Training Experience. Res Q Exerc Sport 2020:1–8. https://doi.org/10.1080/02701367.2019.1679334. We concluded that the maturational effect seemed to be more relevant than the age effect, while body composition seemed to explain better the judokas performance.

Specific comments:

Abstract: 

I suggest replacing the keywords 'young athlete' and 'anthropometry' for 'handgrip strength' and 'adolescence', as the first are also presented in the title.

We appreciate the suggestion. The keywords were changed due to the title change, in response to another reviewer's suggestion.

Results:

Table 1 is the same as Table 1 of Giudicelli et al. (2020). Please, I strongly suggest that the authors provide the differences in the two studies.

The issue has been clarified in response to general comment. Even so, information regarding the normality test was removed from Table 1.

Discussion:

I believe there is a lack of discussion about the explanation of the findings and their application in judo athletes. Why body fat is a negative predictor of several physical tests performance? Which are the implications of this when weight categories is considered? It is known there is a negative relantionship between weight categories and physical tests performance in young (Torres-Luque et al., 2015 SSH) and senior judo athletes (Franchini et al., 2007; Katrali, Goudar, 2012 AJSM; Smulski et al., 2011 JSCR).

Fat mass is a negative predictor of performance in tests that involve displacement of total body mass, such as running and jumping tests, which are among those used in the present investigation. This is mentioned in lines 263-267 (marked file). However, the reviewer's suggestion to discuss body composition considering weight categories was accepted, using one of the suggested references. The inclusion in the text can be seen in the lines 271-278 (marked file).

Is the competitive system in judo (considering age categories) legitimate when the effects of growth and maturation are considered? See Courel-Ibnez, J., Franchini, E., & Escobar-Molina, R. (2018). Is the special judo fitness test index discriminative during formative stages? Age and competitive level differences in U13 and U15 children. Ido Movement for Culture. Journal of Martial Arts Anthropology, 8, 37–41.

We are grateful for the indication of the reference, which gave us more arguments to support the use of indirect tests in this investigation. We have included this issue on lines 287-292 (marked file).

On the question, there still seems to be insufficient knowledge to define whether bio-banding is in fact a viable alternative to chronological age for categorizing young athletes. The discussion seems to be more advanced in football, with unofficial tournaments being organized by federations to test the effects of bio-banding (Cumming SP, Brown DJ, Mitchell S, Bunce J, Hunt D, Hedges C, et al. Premier League academy soccer players’ experiences of competing in a tournament bio-banded for biological maturation. J Sports Sci 2017;36:757–65. https://doi.org/10.1080/02640414.2017.1340656). In judo and other combat sports, studies on the effect of maturation and possibilities of bio-banding are still in their initial stages, so the interest in carrying out the investigation that we submitted for your appreciation.

Other important articles:

Fukuda DH, Stout JR, Kendall KL, et al. The effects of tournament preparation on anthropometric and sport-specific performance measures in youth judo
athletes. Journal of Strength and Conditioning Research 2013;27:331-339.

Fukuda, D. H., Beyer, K. S., Boone, C. H., Wang, R., et al. (2018). Developmental associations with muscle morphology, physical performance, and asymmetry in youth judo athletes. Sport Sciences for Health, 14, 555–562.

I suggest that the authors run the statistical power with the current sample, p-valor and the effect size of the main study variables to determine the statistical power (type II error). Only in case of power <80%, data is limited and unreliable.

We are grateful for the suggestion, and we agree that the calculation of the sample size to control type II error is important for the reliability of inferential analyzes. However, some factors compromise the construction of samples with sufficient N to show statistical power. Investigations carried out with individual sports, including combat sports, unlike research with modalities of collective sports, have difficulty in acquiring larger samples, especially those where the age range is small, as in the present study. Several relevant works in the area, including the kindly suggested articles, have worked with small samples and do not report their statistical power.

As example, Franchini et al. 2007 investigated 22 adult judokas; Courel-Ibnez, J., Franchini, E., & Escobar-Molina, R. (2018) worked with 34; Fukuda et al. 2013 and Fukuda et al. 2018, investigated 20 and 26 judokas, respectively.

To avoid type II error, we used as strategy to take to the hierarchical regression models the anthropometric variables with the highest correlation with the independent variables, which was specified in the text. Furthermore, the discussion and conclusions proposed in the article focused
on the significant effects of inferential analyzes, with little regard for the non-significant effects, which may be subject to the small sample size and, therefore, to type II error.

Conclusion:

The authos concluded that "These findings suggest that bio-banding may be an effective strategy to reduce the implications of maturation on performance, preventing those that are smaller for their age being at a physical disadvantage." Why is the bio-banding method better than other indirect somatic maturation strategies, as peak of height velocity? 

Different maturation indicators can be used as bio-banding strategies. Somatic maturation indicators have been favored over skeletal and sexual maturation indicators due to ethical issues involved (Khamis HJ, Roche AF. Predicting adult stature without using skeletal age: the Khamis-Roche Method. Pediatrics 1994;94:504–7). There is still no consensus to determine which somatic maturation indicator is more suitable to be used as bio-banding. However, predicted mature stature has been used instead of peak of height velocity in studies on bio-banding. The reason is that, although both have correspondence with pubertal status given by stage of pubic hair (Cumming SP, Lloyd RS, Oliver JL, Eisenmann JC, Malina RM. Bio-banding in sport: Applications to competition, talent identification, and strength and conditioning of youth athletes. Strength Cond J 2017;39:34–47. https://doi.org/10.1519/SSC.0000000000000281), the latter seems to have limited application depending on age at the peak of height velocity (Malina RM, Koziel SM. Validation of maturity offset in a longitudinal sample of Polish boys. J Sports Sci 2014;32:424–37. https://doi.org/10.1080/02640414.2014.889846). In this sense, both the predicted adult stature and the peak of height velocity should be the target of research to verify their effectiveness as criteria for bio-banding.

Reviewer 2 Report

Introduction: there is a lack of researches or literature to justify this study. 

As we express our gratitude for your review of our manuscript, we hereby present to the editors and reviewers the changes made to the manuscript, point-by-point, from the suggestions received. While we hope that the corrections made have met the demands arising from the review, we are at your disposal to clarify any doubts and to make any other modifications that you may consider necessary to improve the quality of the manuscript.

There are already several studies in the literature about the effects of biological maturation on the performance of young athletes in team sports, and about the need to classify them with criteria that abdicate chronological age. We intend in the introduction to point out this production, without discussing excessively, since the focus of the work was young judo athletes. Even so, seeking to comply with the suggestion, we changed the text of lines 60-71 and 75-77 (in the marked file), bringing important work on bio-banding in soccer, and better presenting the arguments that justify the investigation. We emphasize that the literature on maturational effect in combat sports and in judo is scarce

Results: Table 1 is huge and many of this data is not relevant. There are some results that have not so much to do with the objective of the study.

In response to the suggestion, the results of the normality tests, previously mentioned in the materials and methods section, were removed from Table 1, and lines 173-176 (in the marked file) were changed to reflect the change

Discussion: Authors should show how relevant are the results and how can be applied, not also for competitions (difficult to change or carry on), maybe also for clubs or trainers. 

To comply with the suggestion, lines 302-304 (marked file) were altered, reinforcing the notion that the application of bio-banding in judo may promote positive sporting experiences, preventing unfair conditions, favoring the maintenance of young athletes in the sport, and the engagement of younger people in sports life.

Reviewer 3 Report

The present study aimed to was to analyze the effects of chronological age, biological maturation, and anthropometric characteristics on the physical performance of young judo athletes. However, numerous problems regarding the research were observed. For example, it seems that the amount of explanations within the text is unbalanced.

As we express our gratitude for your review of our manuscript, we hereby present to the editors and reviewers the changes made to the manuscript, point-by-point, from the suggestions received. While we hope that the corrections made have met the demands arising from the review, we are at your disposal to clarify any doubts and make any other modifications that you may consider necessary to improve the quality of the manuscript

This is a bio-banding manuscript, but not much information was stated in the literature? I felt it is more suitable to report as a fitness test for judo athletes. 

Thanks for the suggestion. Understanding that bio-banding is an important background for the discussion, but not the objective of the study, we changed the title of the article and the keywords

The introduction needs more justification for the purpose of this study.

Academic production on the effect of biological maturation on the performance of young combat sport athletes, specifically in judo, is insufficient. By itself, this already seems to assign relevance to the study. However, we brought in the introduction the available findings on maturation effect in judo, in addition to team sports, where the production is robust. In response to another reviewer, we modified lines 60-71 (in the marked file) of the text, bringing relevant paper that presents the results of the experience in an unofficial soccer competition using bio-banding to distribute athletes in competitive categories, instead of chronological age, as we also reinforced the possible inadequacy of body mass as a criterion for categorizing judo athletes. Please let us know if these changes are sufficient to resolve the issue.

The literature review was insufficient, lack of the latest study.  

We would like to highlight, again, the scarcity of studies dealing with the effects of biological maturation on the training routines and on the performance of combat sports athletes. In addition, the currently production and dissemination of science are fast, making the references used in articles whose main draft is already written, but is still undergoing revision by the various authors, fail to consider more recent references. Despite this, of the 49 references used, 17 are from the last 5 years, (2020-2016), 19 were published between 2015-2011, and 13 were published in 2010 or earlier.
We are open to modify the discussion using more recent texts that we have disregarded. If necessary, please indicate other references that may be relevant.

Table 2, the Pearson correlation, cannot access whether a variable is predictors or not. Wrong analysis. There is no dependent or independent variable in the Pearson correlation. 

Thanks for the comment. Pearson's correlation was not used to identify whether a variable is a predictor of performance or not. With the difficulty of reaching a large sample size, inherent to studies with individual sports athletes, different from what is observed in team sports, the investigation was carried out with a greater number of variables, aiming at greater robustness in the analysis. However, in linear regressions, the number of variables that can be inserted in the model are related to the sample n. Thus, Pearson's correlation was used to determine, among the anthropometric variables, the three most correlated with each physical test, to justify which variables would be inserted in the hierarchical multiple regression models and which would not

The sample size is small (67). Need to determine the power of study with this small sample size. 

We are grateful for the suggestion, and we agree that the calculation of the sample size to control type II error is important for the reliability of inferential analyzes. However, some factors compromise the construction of samples with sufficient N to show statistical power. Investigations carried out with individual sports, including combat sports, unlike research with modalities of collective sports, have difficulty in acquiring larger samples, especially those where the age range is small, as in the present study. Several relevant studies in the area have worked with small samples and do not report their statistical power. As example:

(Franchini E, Nunes AV, Moraes JM, Del Vecchio FB. Physical Fitness and Anthropometrical Profile of the Brazilian Male Judo Team. J Physiol Anthropol 2007;26:59–67. https://doi.org/10.2114/jpa2.26.59) investigated 22 adult judokas.

(Fukuda DH, Beyer KS, Boone CH, Wang R, Monica MB La, Wells AJ, et al. Developmental associations with muscle morphology , physical performance , and asymmetry in youth judo athletes. Sport Sci Health 2018;14:555–62. https://doi.org/10.1007/s11332-018-0460-3) investigated 26 judokas.

(Detanico D, Kons RL, Fukuda DH, Teixeira AS. Physical Performance in Young Judo Athletes: Influence of Somatic Maturation, Growth and Training Experience. Res Q Exerc Sport 2020:1–8. https://doi.org/10.1080/02701367.2019.1679334) investigated 66 young judokas

(Courel-Ibnez J, Franchini E, Escobar-Molina R. Is the Special Judo Fitness Test Index discriminative during formative stages? Age and competitive level differences in U13 and U15 children. Ido Mov Cult Martial Arts Anthropol 2018;18:37–41. https://doi.org/10.14589/ido.18.3.6) investigated 34 judokas

To avoid type II error, we used as a strategy to take to the hierarchical regression models the anthropometric variables with the highest correlation with the independent variables, which was specified in the text. Furthermore, the discussion and conclusions proposed in the article focused on the significant effects of inferential analyzes, with little regard for the non-significant effects, which may be subject to the small sample size and, therefore, to type II error.

The Methods section is quite a confusion, and it should be divided into several sub-sections. 

We are grateful for the recommendation. The Materials and Methods section was prepared based on the template provided by the journal (available at https://www.mdpi.com/journal/ijerph/instructions). However, we accept the suggestion and divided the section into the following subsections: participants; anthropometric measures; biological maturation; physical performance; procedures; and statistical analysis

Authors did not explain the study design and sampling method used in the study. 

In addition to the information already described in the Materials and Methods section, information about the study design and the sample selection were included in line 87 (marked file).

If this is a cross-sectional study, the multiple linear regression will not give the answer on causal effect; therefore, the independent variables should not be called as predictor variables. 

Thanks for the explanation. The misuse of the term seems to be common in the literature, and we ended up repeating it. The entire text of the paper was changed to correct the terminology

I felt the study need stronger justification and a stronger conceptual framework.

The wording of the article was considerably modified to meet the suggestions and comments of the three reviewers. We hope that the modifications made will be sufficient to meet the journal's publication standard, at the time we are available to exchange more information and to modify the text further, if necessary.

Unfortunately, I feel that the current manuscript was not sufficient to read the required level for publication. 

Round 2

Reviewer 1 Report

The authors provide satisfactory answers about my questions, especially regarding the data and previous study. In addition, improvements in the quality of the text were provided.

As we express our gratitude for your review of our manuscript, we hereby present to the editors and reviewers the changes made to the manuscript, point-by-point, from the suggestions received. While we hope that the corrections made have met the demands arising from the review, we are at your disposal to clarify any doubts and make any other modifications that you may consider necessary to improve the quality of the manuscript.

I have just a few questions. A brief explanation of use of predicted mature stature instead of other methods, such as peak of height velocity may be a complement to the practical applications, since PHV is widely used, but it has limitations, which makes PMS more interesting.

Thanks for the suggestion. To address it, in addition to the explanation already offered in the section "materials and methods", we insert a paragraph at the end of the discussion section (lines 282-289 of the new version of the text), with one more reference, reinforcing the limitations of the PHV for bio-banding and highlighting that the literature on the subject has mostly used the predicted adult stature. Thus, we preferred to use PAS to contribute to the literature and to allow the present study to be objectively compared to other investigations.

Some references are in the list, but were not cited in the text. For example, the references included (43 and 48) were not cited in the text. The reference 47 is also not cited, but I'm not sure if it should be maintained.

Thank you for drawing attention to the problem. It was probably due to the insertion of new references to the text using Mendeley while Word's change control function was activated. I have reviewed all the references and I believe they are all appropriate now. Regarding reference 47, we accept the suggestion and remove it from the text.

Reviewer 2 Report

Thanks for considering suggestions. I think it has been improved.

Reviewer 3 Report

Authors had attempted to address the comments as suggested with satisfaction. 
